# The Cosmic Zoo: The (Near) Inevitability of the Evolution of Complex, Macroscopic Life

**DOI:** 10.3390/life6030025

**Published:** 2016-06-30

**Authors:** William Bains, Dirk Schulze-Makuch

**Affiliations:** 1Department of Earth, Atmospheric and Planetary Science, MIT, 77 Mass. Ave., Cambridge, MA 02139, USA; 2Rufus Scientific Ltd., 37 The Moor, Melbourn, Royston, Herts SG8 6ED, UK; 3School of the Environment, Washington State University, Pullman, WA 99164, USA; dirksm@wsu.edu; 4Center for Astronomy and Astrophysics, Technical University Berlin, Berlin 10623, Germany

**Keywords:** transition, key innovation, complexity, evolution, multicellularity, origin of life, great filter

## Abstract

Life on Earth provides a unique biological record from single-cell microbes to technologically intelligent life forms. Our evolution is marked by several major steps or innovations along a path of increasing complexity from microbes to space-faring humans. Here we identify various major key innovations, and use an analytical toolset consisting of a set of models to analyse how likely each key innovation is to occur. Our conclusion is that once the origin of life is accomplished, most of the key innovations can occur rather readily. The conclusion for other worlds is that if the origin of life can occur rather easily, we should live in a cosmic zoo, as the innovations necessary to lead to complex life will occur with high probability given sufficient time and habitat. On the other hand, if the origin of life is rare, then we might live in a rather empty universe.

## 1. Introduction

All discussion and teaching on the evolution of humans has embedded in it the idea of key innovations on the path from non-living matter to humanity. We can identify features of our current biology that are central to our existence and nature, such as aerobic respiration, and those that are peripheral, such as possession of earlobes, and seek to find the point in our evolution where the key features were acquired. Debate on this in recent times has been re-ignited by Smith and Szathmary [1]. In this paper, we analyse the transitions or key innovations within a theoretical framework that allows us to ask whether the evolution of a technology-using species on Earth is an extremely unlikely event, or whether complex, smart and potentially technological beings are highly likely to evolve on an habitable planet in the time available to it (10 Gigayears (Ga) in the case of the Earth orbiting around our Sun).

This is a highly anthropocentric approach, but we take it deliberately because we are interested in the evolution of complex and intelligent organisms such as ourselves (McShea and Simpson [2]). No evidence of technologically advanced life other than human life has been found, which suggests that such technologically advanced life occurs only on a minor fraction of all habitable planets. An important question is therefore whether there exists what Robin Hanson calls “The Great Filter” somewhere between the formation of planets and the rise of technological civilizations [3,4]. That filter could, in principle, be any of the many steps that have led to modern humanity over the last 3.8 Ga [5,6,7]. Which of these major key innovations are highly likely, which are unlikely? Where does the “Great Filter” lie? 

There are two clarifications we need to emphasize before we can ask whether the evolution of complex life is likely. The first is what we mean by “complex”. It is notoriously difficult to define complexity [8]. Complexity can be related to the information content of a system, or the amount of information needed to specify it (although that itself depends on how the specification is done) [9]; under such information-based complexity (IBC) theory, by any measure that humans require more information to describe them than *Escherichia coli* (*E. coli*), and hence are more complex. Even the simplest living thing is very complicated compared to inanimate matter. However, it is clear that a fruit fly has more types of cells, more complex cells, more interactions between those cells, a wider range of behaviours and a more complex genetic program to bring all that about, all of which are required for its survival, than an equivalent mass of *E. coli*. The mass of *E. coli* can be described by describing one *E. coli* and then saying “grow 10^9^ of them”, describing a fruit fly requires describing all of its cell types and their interactions in chemistry, space and time. Thus, when we discuss complex life we generally mean macroscopic multicellular life, with our goal being to review the evolution of function in life’s history toward achieving higher levels of complexity.

Only a rather small fraction of Earth’s biological endowment has achieved a high level of complexity. Most of the biosphere remains genetically, morphologically, and behaviourally quite simple, which is the second clarification needed [10]. Conway Morris characterises the apparent increase in biological complexity with the aphorism, “Once there were bacteria, now there is New York” [11], but we should not forget that there are still bacteria, and more of them than New York (or New Yorkers). Probably around 50% of the biosphere by mass and by genome number is bacterial or archaeal [12,13]. Of the eukaryotes, the majority by number and by genetic diversity are unicellular. As discussed below, only three clades of eukaryotes have evolved true multicellularity. The Animalia are a tiny fraction of all life, and humans are just one animal. We recognise that focusing on humans is a highly distorted perspective on evolution in general, and we do not wish to imply that the evolution of complex, intelligent life is inevitable because of some directionality in evolution.

Our argument rather is that the evolution of complex life is likely. This rests on two arguments. The first is that the functions found in complex organisms have evolved multiple times, an argument we will elaborate in the bulk of this paper. The second is what Gould calls “Diffusion from the Wall” [14]. There is a limit of complexity below which life cannot function (see e.g., [15,16]). It seems plausible that life started as a simple organism, close to this “wall” of minimum complexity [17]. From that simple Last Common Ancestor (LCA), life can evolve genetic, morphological, developmental or behavioural complexity in one of three directions. It can become simpler, it can remain the same, or it can become more complex. If the LCA was a “minimal cell” then it cannot become simpler. However it can become more complex. Such more complex life can also evolve to become simpler or more complex. With time, the most complex life (however complexity is defined) is therefore likely to become more complex. While evolution of simpler forms from complex ones is common, and while the “average complexity” of the biosphere might be unchanged (if it is meaningful at all), the most complex organisms are likely to be more complex.

## 2. Hypothesis

Our hypothesis is that the evolution of complex life is highly likely in any stable, sufficiently extensive environment where there is life. By “complex life” we are specifically interested in obligate multicellular life-forms, particularly members of the kingdoms Plantae (plants), Fungi, and Animalia (animals). If the Great Filter is at the origin of life, we live in a relatively empty universe, but if the origin of life is common, we live in a Cosmic Zoo where such complex life is abundant. Deeper understanding of life on our planet with all of its diversity will provide indications whether our hypothesis is correct. The final test though will be when our space faring and remote sensing abilities allow us to explore planets and moons beyond our solar system and possible biospheres on them.

## 3. Key Innovations of Life: Background and Models

### 3.1. Background

Discussions of key innovations of life usually describe events in the paleontological record that represent an innovation that qualitatively changes the nature of the biosphere, and in particular the development of a new type of chemistry, morphology or inferred behaviour. These represent new capabilities, and not only quantitative expansion of pre-existing life-forms, although of course the existing life-forms continue, and (as noted above) may continue to dominate the biosphere. Originally the term “Key Innovation” was applied to steps that allowed the radiation of particular groups of organisms [18,19], but recently the concept has also been applied to higher taxonomic levels, and hence to more fundamental changes in the nature of the biosphere. There is some unanimity in what these high level, major steps to human life are. We have summarised these in Table 1, which emphasizes the functional definition of a key innovation. We are not arguing for what we will here parody as The Star Trek Fallacy—that it is inevitable that complex, intelligent aliens will all have pentadactyl limbs, circular irises and male-restricted facial hair. With Stephen J. Gould [20], and in some disagreement with Conway-Morris [21], we agree that rewinding and replaying the tape of life will not result in humans. Here we are concerned with the processes that result in function, not specific anatomy. The vertebrate eye is specific, but vision has evolved many times [22].

Our approach is explicitly anthropocentric, because it focuses on the path from the origin of life to humans. Gould [14] has criticised this anthropocentric approach, pointing out that “primitive” microorganisms dominate the Earth by numbers, mass and chemistry. This is true, but not relevant, for two reasons. Firstly, our goal is to explore the path to complex life, not to put that path in context of the biosphere in general. Secondly, as we will discuss below, evolution of a large and diverse microbial biosphere is a prerequisite for evolution of complex organisms that are at higher trophic levels. That microorganisms have adapted to quite extreme environments is interesting, but has little impact on either of these end points.

### 3.2. Models

Many authors (e.g., [1,23,24,25,26,27,28] have previously argued that the evolution of life on Earth occurred in major steps and several of these major steps or key innovations have been identified. Examples are the invention of photosynthesis, eukaryotic gene organization, the advent of multicellular life, among others (Table 1). These major innovations have occurred in a sequence spread out over geological time. Explaining the history of how they occur is a huge, and incomplete, task. Even the simplest biological systems are highly intricate and highly diverse, and there is no way even in principle to predict their evolution; in Kauffman’s terms [29] life is non-ergodic, and which of the vast number of possible organisms actually happens is therefore a matter of chance and history [30]. Trying to say anything general about the evolution of life as a whole is therefore futile in the absence of some simplification. 

The mechanism by which life acquires a new function is also arbitrary and inherently unknowable in advance [29]. Jacob famously commented that evolution is a “tinkerer”, taking existing structure and using small changes to adapt it to new uses [31]. This means that how a new function is acquired depends both on the panoply of structures that do not have that function in the ancestor organism and the specifics of the mutations that allow one of those structures to acquire a new function. Both are unknowable today. 

We chose to simplify by focusing on outputs and function. Rather than asking about the details of how an evolutionary step leading to a new function has happened, or the ancestry of a specific mechanism (which is inevitably specific to the organism concerned), we ask whether it has happened. That at least should be uncontroversial. By analogy, predicting how a flipped coin will fall is enormously complex, requiring knowledge of the initial position, velocity and angular momentum, air resistance, inelastic properties of the surface, mechanical properties of the coin and so on. However, if we observe many coin flips, we can estimate how likely a “heads” is to come up. The fact that (unlike coin flips) the space of possible biological outcomes in unknowable [29] is not relevant. In the evolutionary coins that have been flipped to date, we observe (for example) imaging vision occurring at least 10 times [22], and conclude that its evolution is therefore likely (On a planet in which there is visible light at the surface and there are organisms with sizes greater than the wavelength of light. For the case of a perpetually dark world or one inhabited only by micron-sized organisms, we might review our estimate). 

In this framework, we have suggested [32] that there are three major paths to innovation, based on the occurrence of chance in those paths, which can be summarized briefly as follows:
A Critical Path Model. The major event or innovation requires preconditions that take time to develop. The amount of time needed is to a large degree determined by the nature of the event and the geological and environmental conditions of the planet, and so once the necessary preconditions exist on the planet then the event will occur in a well-defined time frame. There is no major role of chance in this process.A Random Walk Model. The major event or innovation is highly unlikely to occur in a specific time step, and the likelihood does not change (substantially) with time. This may be because the innovation requires a highly improbable precursor event to occur, or a number of highly improbable steps that have to occur in sequence. Thus, a substantial amount of time has to elapse before chance events allow the innovation to be made. Once life exists on a planet, ultimately the innovation will occur, but when it occurs is up to chance, and whether it occurs before the planet’s sun leaves the main sequence and renders the planet uninhabitable is not knowable. Chance events limit this process, and must occur in a specific order.A Many Paths Model. The major event or innovation requires many random events to create a complex new function, but many combinations of these can generate the same functional output, even though the genetic or anatomical details of the different outputs are not the same. So once life exists the chance that the innovation will occur in a given time period is high, but the exact time is not knowable. Chance events limit this process, but can occur in any order.

Each of these may also fall into a fourth category, termed as “Pulling Up the Ladder”. Either a transition leaves the world in which it occurred relatively unchanged with respect to the chance of that transition (our default assumption), or it changes the world with respect to the chance of that transition. In this latter class of explanation, which we call a “Pulling Up the Ladder” process, an innovation is likely (either because it is a Critical Path or a Many Paths process), but the results of the innovation destroy the preconditions for its own occurrence. The new organisms “pull up the ladder after themselves”. The endosymbiotic origin of plastids could be a “pulling up the ladder” process, because once the plant ancestor had acquired a proto-chloroplast, there was no opportunity for it to acquire another, equivalent endosymbiont. 

The reason for classifying innovations in this way is that the three classes of models have different implications for the likely timing of the events. Evolutionary events are inherently random, and so cannot be predicted. However, the chance of a repeat of an innovation can be estimated from a model of what is needed for that innovation to occur, just as the chance of any random event can be estimated from knowledge of the mechanisms behind the event. We have suggested that the three classes of model have different implications for the chance that a function will evolve within a time period.
Critical Path Model. One set of preconditions is needed for that transition. Once those preconditions (“causes”) are satisfied, the innovation will arise quickly, and will occur on all occasions that the preconditions are satisfied. The preconditions take only time to fulfil, and there is no (major) random element in it (however, the required amount of time may be very substantial). As a consequence, if an innovation occurs through a Critical Path process more than once, it is likely to follow a similar evolutionary path in the different examples. Thus, independent evolution of a common function in the descendants of a common ancestor is likely to use similar mechanisms.Random Walk Model. There are no preconditions other that prior existence of life that can achieve the innovation (e.g., nervous systems cannot evolve without cells). The innovation will occur at random, but since it is highly improbable it will not likely occur twice even if the preconditions are satisfied many times.Many Paths Model. There are no specific preconditions other that prior existence of life that can achieve the innovation. However once any appropriate precondition is met, the innovation will occur at a fairly reliable time frame (in generations) afterwards, and so will eventually occur on all occasions that the preconditions are satisfied. If an innovation occurs through a Many Paths process more than once, it is likely to use different mechanisms each time it occurs.

## 4. Preconditions for Complex Life

There are planetary conditions that have to exist for life to be possible. The planet in question has to be habitable, meaning that the planet provides the minimum shelter and resource requirements for life to thrive. Irwin and Schulze-Makuch [33] identified these requirements as access to energy, complex chemistry, habitat availability, and solvents that are present in liquid form. Kasting [34] proposed the concept of the “circumstellar habitable zone” (HZ), the zone in which an Earth-like planet would retain an atmosphere and liquid water. This zone is quite narrow for most stars and led to the development of the “Rare Earth” argument by Ward and Brownlee [35] that complex (meaning non-microbial) life is rare in the universe. Conway-Morris also considers complex life, specifically human life, unlikely [21]. The rarity of habitable planets based on the HZ concept has been questioned on two grounds. Firstly, the chemistry of life could be different from that of Earth, and specifically could not necessarily require liquid water [26,36,37]. Secondly, non-Earth-like planets could have surface liquid water over a much wider range of orbital parameters [38,39]. 

Schulze-Makuch and others [40] developed the habitability concept further and proposed a Planetary Habitability Index (PHI), aimed for exoplanets, based on a non-Earth-centric approach. Evaluation parameters included the presence of a stable substrate, available energy, appropriate chemistry, and the potential for holding a liquid solvent. To develop a sustainable biosphere, the planet or moon has to be terrestrial (rocky surface) and have an atmosphere or at least an ice cover to keep the water (or other solvent) liquid and protect it from radiation. Beneficial parameters were the presence of a magnetic field, internal differentiation of the planetary body and plate tectonics as a particularly efficient recycling mechanism. To arrive at a complex biosphere with multicellular, macroscopic life, perhaps even technologically advanced life, time also is needed. Earth serves as a pointed example: while life on Earth likely originated within the first 700 Ma (million years, Ma) of the surface becoming habitable, it took at least another 2 Ga until multicellular, macroscopic life evolved, and only now, after more than 4.5 Ga of earth’s natural history, a species arose that we would call technologically advanced. This time dependency was realized by Irwin et al. [41] when they developed a so-called Biological Complexity Index (BCI), to relate the likelihood of the development of more complex life on a planetary body
BCI = (T G A)^1/3^(1)
where T represents the temperature, G the geophysical attributes (density plus orbital properties), and A the age of the planetary body, respectively. 

Terrestrial life can flourish between −20 °C and 120 °C, although eukaryotic life seems limited to between about −10 °C and 70 °C [42]. The geophysical attributes apply to all life. The age or minimum time it takes to develop biological complexity is unknown, but some limits can be estimated. If Darwinian evolution requires a large amount of time to explore the “genetic space” of solutions to the hurdles and transitions explored in more detail below, the planetary body in question would have to be habitable not only for a long time period, and for some or most of that time provide abundant energy. Oxygenation of Earth’s atmosphere may have been a prerequisite for the development of complex macroscopic life such as animals [43], and if so, it is hard to see how this can be accomplished on a terrestrial planet in less than 1 Ga. However, we have to be careful to not categorically exclude any possible shortcut solution to the required time scale. 

The habitability of planets change with time. Earth has gone through at least two “Snowball” events in which the large majority of the surface was uninhabitable, and a number of other mass extinctions in which the large majority of then extant genera became extinct. Such events are random with respect to the complexity of life. However, they do not set back the evolution of the biosphere substantially. We know little about the recovery life after the paleoproterozoic snowball [44] except that the then newly evolved oxygenic photosynthesisers survived the event, but the Cryogenian snowball is argued to have accelerated the development of multicellular life [45], and recovery of flora and fauna after Phanerozoic mass extinctions has been relatively rapid (5–20 Matimescale) with no overall loss of diversity or complexity (see for example [46,47,48,49]). 

Of course, it is possible for a geological change to be so extreme that the planet’s surface becomes permanently uninhabitable, as has happened to Mars [50,51] and possibly Venus [52]. If the Snowball Earth had never melted, then the habitability criteria above would not be fulfilled.

Given these constraints, what are the chances that life arises and develops animals as complex as humans?

## 5. The Key Innovations

### 5.1. Origin of Life

Despite nearly 150 years since Darwin speculated on life’s appearance in “some warm little pond” in his letter to Hooker, J.D. on 1 February 1871, we do not know how life appeared on Earth [53]. We therefore cannot say whether the origin of life was an extremely unlikely singular event or whether it happened many times. All terrestrial life shares the same underlying biochemistry: is this because this biochemistry was the first to appear (which then consumed all organic matter, removing the possibility of a second, independent origin for life—a Pulling Up the Ladder event), that our biochemistry is the best fitted (and so independent origins of life converged on it), or that it is the frozen result of an extremely unlikely event? 

Life was definitely established on Earth 3.5 Ga ago, and may have been widespread as early as 3.8 Ga ago, only 50 to 100 Ma after the end of the Late Heavy Bombardment [54,55,56]. We can be confident that the Earth was sterile immediately after the proposed Moon-forming impact ~4.5 Ga ago [57,58,59]. Therefore life arose on Earth within a 700 Ma timeframe. However this is a weak constraint; does this represent an average for terrestrial planets, or was Earth incredibly lucky? With only one example of the origin of life, we cannot tell.

We therefore have to examine the possible paths to the origin of life, to see if we can identify Random walk or Many Paths events in that path. This is unsatisfactory given our philosophy as outlined above, but in the absence of finding an independently originated lifeform on another world, historical reconstruction is all that is left to us. As we shall see, it does not answer our question. 

The origin of life (OOL) event included the initial appearance of a coded, bounded replicator and the acquisition of the chemical complexity necessary in the Last Common Ancestor (LCA). Many laboratory models of prebiotic conditions have shown that quite complicated organic molecules can be made under vaguely plausible prebiotic conditions [60]. The discovery of amino acids, alcohols, trioses, and other biologically relevant molecules in meteorites and in interstellar space attest to the ease with which abiological chemistry can make them. A range of experiments shows some limited patterned replication of proteinoids, nucleic acid-like molecules, and larger structures such as miscelles in completely abiological systems [61,62]. Thus complicated chemistry clearly can happen. Other schemes show how geochemical energy flux can be coupled to generate such chemistry, and power it (e.g., [60]).

However these chemical demonstrations do not address whether the origin of life as a complete process could occur many times. Life needs to have catalysis, coding and containment. Patterned replication is not sufficient, chemistry without containment cannot be the basis for life, and macromolecular catalysts coded by the genetic system are needed both to catalyse the chemistry of the system and couple external energy gradients into that chemistry. All of these components need to come together at the same place and time to create life. Once coded replication is achieved, Darwinian evolution becomes a logical inevitability [30] and life is able to pursue all the steps below. Until coding is achieved, the abiological chemistry, no matter how complex, is just organic molecules approaching equilibrium conditions with a minimum value of internal energy. 

We do not know at the moment in what order the components of life appeared on Earth—whether informational macromolecules, metabolic activity, containment [63,64] or a combination of these was precedent. For our purpose, we can just note that if the functions of life have to appear in a specific order (no matter what that order is), then it is more likely that the origin of life is a Random Walk event. If by contrast the functions of life can appear in different orders, such that a primitive metabolism could drive the formation of a genetic apparatus or a primitive genetic apparatus could organize a metabolism, then the origin of life may be a Many Paths process, and consequently more likely.

However, at the moment, we have no way of deciding what category of explanation is appropriate for the origin of life, nor of the relative probability of it occurring elsewhere. The relatively early occurrence of life on Earth may suggest it is a high-probability event, which only occurred once because life itself would consume its precursors, precluding a second origin of life (a “Pulling Up the ladder” process). Or the initial development of a primitive genetic code may be an extremely unlikely Random Walk event, which on Earth occurred rapidly (or, under some hypotheses, occurred elsewhere and then was transported to Earth), but elsewhere may occur only after billions of years, or not at all. The origin of life remains an unknown in our analysis.

### 5.2. Photosynthesis

All hypotheses on the origin of life imply that life originally obtained energy from geological sources [65]. The ability to capture the energy of light into chemical synthesis allowed life to transcend local geochemical energy sources. Light is a stable, abundant, dense source of chemical energy, readily available on the surface of any terrestrial planet. How often did that ability appear? There are several lines of argument that suggest this complex process follows the Many Paths model, and hence is inherently likely to occur. Rothschild [65] comes to a similar conclusion, based on the common presence of light and inorganic carbon on any habitable exoplanet, and the independent, diverse chemistries used to fix carbon on Earth (discussed below). 

The vast majority of photosynthesis on Earth relies on chlorophyll-based photon capture mechanisms. These have a common evolutionary origin in the metabolic pathways for chlorophyll [66]. Chlorophyll-based photosynthesis is found in five different bacterial phyla (reviewed in [67]), but this is probably a consequence of horizontal gene transfer [68] and chlorophyll synthesis only evolved once [69]. There is robust evidence that chlorophyll-based photosynthesis evolved before oxygenesis, not least of which is that primitive chlorophyll biosynthetic pathways are oxygen intolerant [66]. Chlorophyll-based photosynthesis evolved “fast”, and hence might not be an unlikely evolutionary innovation. Two types of chlorophyll-based photosynthesis (PS-I- and PS-II-like systems) in bacteria probably evolved by duplication and divergence of a common ancestral system [70]. 

Other components of the photosynthetic apparatus, such as the antenna complexes, differ substantially between major groups of organisms, and probably were co-opted from different genes and pigments in different organisms [66], i.e., their appearance followed the Multiple Path model.

However, chlorophyll-based photosynthesis is not the only mechanism for the capture of photon energy into chemical energy. Bacteriorhodopsin-based photon capture is chemically completely different from chlorophyll-based capture [71,72]. Most bacteriorhodopsins are sensory, but the more recently discovered proteorhodopsin forms the core of a non-chlorophyll-based light-energy capture system in a wide range of marine prokaryotes [73,74], as well as providing supplementary energy supplies for some halobacteria [75]. Bacteriorhodopsin uses retinal as its photon-absorbing pigment [72], and transduces the energy of a photon to translocate a proton, generating a membrane potential that can then be coupled to chemical synthesis. Only ~300 mV can be generated, so this is not a mechanism that (in its present form) could power oxygenesis, but it nevertheless plays a major role in the marine ecosystem. The marine bacterioplankton clade SAR11 (the smallest free-living organism currently known) photosynthesises using proteorhodopsin [76], although it seems to use this to supplement heterotrophic energy capture, and is therefore not fully autotrophic. These primarily prokaryotic phototrophic proteins have been acquired by dinoflagellate eukaryotes at least twice by horizontal gene transfer, and at least once to form the basis of functional intracellular energy-generating structures. There are three other photopigment systems other than chlorophyll and retinals used in living systems, although probably not to capture substantial metabolic energy [65,77]. 

In photosynthesis, the origination of light-capture chemistry is hypothesised to derive from the chemistry that protected early organisms from UV light damage. The UV protective mechanism had to be able to absorb UV photons and isolate the resulting triplet excited states until they could decay through thermal decay or transfer the excitation energy to another, unreactive molecule [78]. This was achieved by a combination of pigment and “protecting” protein. There are modern examples of organisms that have coupled photon protection mechanisms to energy capture, and which can be regarded as “missing links” between UV-tolerant organisms and ones capable of efficient photosynthesis. Melanin in melanised fungi can generate electrochemical gradients in response to ionizing radiation [79,80] and possibly UV and visible light [81], although very inefficiently compared to bacteriorhodopsin- or chlorophyll-based systems. This has been claimed to be coupled in vivo to ATP synthesis. The critical step of protecting the organism from destructive triplet states and transferring energy to other molecules has occurred here. 

Valmalette et al., identified that the aphid *Acyrthosiphon pisum*, whose genome encodes catotenoid production [82], can use those carotenoids to capture light energy and use this to drive electron transport and ATP production in mitochondria [83]. This is a different mechanism of light energy capture from other discussed above, and so represents a fourth instance of the independent evolution of photosynthesis.

Thus photon energy capture evolved independently at least twice, and arguably four times, and so we argue that photosynthesis represents a Many Paths process. The capture of photon energy into chemical energy, a core step in photosynthesis, is a Many Paths key innovation. 

Photosynthesis is primarily useful for providing energy for the reduction of environmental carbon [84]. There is ample evidence that the carbon-capture components of the photosynthetic biochemistry show many evolutionary routes. There are six known pathways for fixing carbon dioxide, of which the Calvin Cycle used in oxygenic phototrophs is the least efficient in terms of the energy and the reducing equivalents (electrons) required per mole of fixed CO_2_ [85]. The carbon isotope ratio at 3.5 Ga ago is interpreted as evidence that microbial ribulose-1,5-biphosphate carboxylase (RuBisCo)-based carbon fixation occurred then [86]. Even within specific reactions, independent evolution is known. For example, diatom carbonic anhydrases are apparently unrelated to any others, and some marine picophytoplankton have no carbonic anhydrase, suggesting different mechanism of the initial steps in CO_2_ capture [87]. Rothschild [65] lists 23 carboxylase enzymes that can be used to capture CO_2_ into organic molecules (obviously, most are not part of net carbon fixation). Thus the accessory reactions of photosynthesis, including the “Dark Reactions” of plant photosynthesis, could have evolved through a Many Paths process as well.

### 5.3. Oxygenesis

Oxygenesis is regarded as being of central importance to the development of complex life. For example, Canfield [88] stated that “The evolution of oxygen-producing cyanobacteria was arguably the most significant event in the history of life after the evolution of life itself”. Thus many studies on the evolution of photosynthesis usually focus on the evolution of oxygenic photosynthesis (see reviews in [66,89]).

Most photosynthetic life on today’s Earth (by mass) uses water as an electron donor, generating molecular oxygen as a waste product [65]. This is despite the substantial drawbacks of oxygenesis. Oxygenic photosynthesis requires significantly more investment in cellular machinery and its genes than anoxygenic photosynthesis (and hence slower growth rates, all other things being equal [90]). Furthermore, the molecular oxygen and the reactive oxygen species it can generate are inherently damaging and dangerous chemical species [91].

However, the great advantage provided by oxygenesis was its capacity to liberate life from the need to find rare electron donors such as sulphide, hydrogen or Fe(II) to support the reduction of carbon dioxide, giving oxygenic photosynthesisers an advantage over all other forms of life (e.g., [44]). The thick carbonate deposits in the 10–50 Ma before the Great Oxygenation Event at ~2.4 Ga ago support the idea that life had made a major productivity breakthrough at that time. However, it is not clear that this advantage was true for the earliest oxygenic photosynthesisers [70]. Two equally plausible explanations for the evolution of oxygenesis are allelopathy, the release of one or more biochemicals from one organism to affect growth, survival or reproduction of another one, and land colonization. The earliest oxygenic organisms may have evolved from land soil actinobacteria, living in a fresh water [92], high-UV environment that would have been extremely poor in sulphide and iron [93], an environment that would strongly favour an organism that could retask UV-protection mechanisms to splitting water [70,94]. They may also have used oxygen both as a waste product from electron abstraction and for allelopathy, blocking other photosynthesizing organisms in their immediate vicinity from using sulphide or ferrous iron as a source of electrons, a strategy that works best for a minor species competing in a densely populated niche. Oxygen may also have been valuable in oxidizing, and hence mobilizing, Mo and V necessary for nitrogenase function [95].

Oxygenesis had clearly evolved before 2.4 Ga years (reviewed in [96]) when oxygen became a significant component of our atmosphere, leaving hopanoid biomarkers of cyanobacteria in rocks [66,70,97,98]. Several indirect lines of evidence suggest oxygenesis could have occurred at least locally as far back as 3.5 Ga ago [66,88,97]. Despite this antiquity, all oxygenic photosynthesisers use the same molecular mechanisms to capture light energy and split water, speaking to a single occurrence of this key step, and only one clade of organisms has evolved this capability [66,99].

Because oxygenesis only evolved once, we are forced again to ask what we know about its evolution, with the goal of seeing whether sub-steps in that evolution are themselves Many Paths events. As with the Origin of Life, this is unsatisfactory as it undermines our approach, but for completeness we summarise current thinking on the evolution of oxygenesis here so as to rule out a clear Many Paths process. 

There is a well-established hypothesis for how the photon-capture mechanism of oxygenic photosynthesis evolved by duplication, specialization and coupling of a simpler, non-oxygenic photosynthetic system to generate the two-centre system of today [78]. Evolution of highly oxidizing PS-II through duplication [100] is still disputed in regard which reaction centre is more like the original (see e.g., [78,100]), but for our purposes this is not important.

The chemical challenges to evolving this metabolic pathway are:
that two water molecules must be oxidized to produce one molecule of O_2_, while at the same time dispensing four charge separated electron/proton pairs [60];PS-II must be shifted to a strong positive oxidizing potential while PS-I is shifted to a highly negative one [99]; andthe oxygen sensitive components of an anoxygenic photosynthetic apparatus (especially the FeS clusters) must be transformed into oxygen resistant ones [101].

There are several hypotheses on how the transition has occurred, which mostly centre on the manganese cluster needed to catalyse the reaction (see e.g., [99,102,103]), while some hypotheses also postulate intermediate electron donors such as hydrogen peroxide [99], or Mn(II) [104,105]. 

Oxygenesis evolved only once. There are two possible explanations for this. One is that it is a Random Walk process, requiring a sequence of unlikely evolutionary steps, which would not have evolved elsewhere. The hypotheses on the origins of oxygenesis above hint this may not be the case, but do not prove it. The other explanation is that the evolution of oxygenesis is a Many Paths process, one which has a high probability of occurring, but is also a Pulling Up the Ladder event, such that once oxygenesis evolved once that evolution removed the preconditions for its evolution again, in this case filling the niche of a photosynthesiser freed from limitation of an electron donor supply. The biochemistry of oxygenic photosynthesis points toward this second explanation. There are six known pathways for fixing atmospheric carbon, of which the Calvin Cycle used in oxygenic phototrophs is the least efficient in terms of the energy and the reducing equivalents (electrons) required per mole of fixed CO_2_ [85]. Rubisco has a very low turnover for fixing carbon, and its carboxylase activity is compromised by opposing oxygenase activity that uses molecular oxygen to break down Ribulose-1,5-bisphosphate rather than fix CO_2_ into it [106]. Despite this, the first inventor of water-splitting was successful, and filled the niche.

The evolution of oxygenic photosynthesis occurred early in life’s history, arising from a precursor (anoxygenic photosynthesis), which has arisen several times, and once arisen oxygenesis removed the drive to evolve a different alternative. The fact that it arose early in the history of life and has been adopted almost universally throughout the eukaryotic domain, is consistent with a Many Paths process leading to a Pulling Up the Ladder event. However, this is a weak argument, the single evolution of oxygenesis is also consistent with a Random Walk event, and the possibility exists that the evolution of oxygenesis may be a critical, improbable step on the path to complex life. 

### 5.4. Endosymbiosis and Eukaryotic Cell Structure

The invention of the eukaryotic cell, together with the development of eukaryotic gene organization, has to be considered a key transition on the path to complex organisms. We are concerned ultimately with the development of obligately multicellular organisms, i.e., organisms in which many cells differentiate into different, specialist functions, and such differentiation is essential for survival. While not all eukaryotes are multicellular organisms, all obligately multicellular organisms are eukaryotes. Many cyanobacteria grow as multicellular filaments, but in all cases these only show differentiation into more than one cell type when environmental stresses trigger cell differentiation [107,108]: filaments of only one cell type are common, and can be broken into arbitrarily small fragments (including one cell) in normal environments and remain viable [109].

Robust evidence for eukaryotes exists starting from 1.9 Ga and 1.7 Ga ago [110]. Thus, eukaryotes appeared a considerable time after the advent of the prokaryotic cell, possibly up to 2 Ga later. It is widely accepted that the modern eukaryotic cell evolved by a series of endosymbiotic events [111,112]. Chimeric models indicate that the first eukaryotic organism originated by the merging of an archaea and a bacterium, either by accidentally joined cells (physical fusion, see also the section below on Multicellularity) or by endosymbiosis (reviewed in [113]). It has also been proposed that some other organelles such as cilia, flagella, centrioles, and microtubules evolved through endosymbiosis, although these hypotheses are highly controversial. 

The origin of the eukaryotic cell is believed to be intrinsically related to endosymbiosis and involved compartmentalised cells with specialist functions carried out by distinct organelles allowing more efficient scaling of cells to larger size and hence more complexity [109]. Endosymbiosis requires resolution of genetic conflicts between the two cell lineages. This is an envisioned difficulty, which is why Blackstone [114] argued that eukaryotic cells may have only originated once. If this is so, it can be understood that the first fusion or endosymbiosis with the result of a eukaryotic cell was a pull-up-the-ladder event, so successful and occurring so rarely that it changed the biota very quickly. 

However, biology does not support this. Endosymbiosis is a common and re-occurring theme in the evolution of life. Endosymbiosis can be observed in many other instances such as from nitrogen-fixing bacteria in root nodules (e.g., *Rhicobium leguminosarum*), single-cell algae inside reef-building corals, and bacterial endosymbionts that provide essential nutrients to about 10%–15% of all insects. Some organisms that do not have mitochondria (e.g., the amoeba *Pelomyxa* or the protozoa *Mixotricha paradoxa*) have aerobic bacteria as symbionts that provide a similar function as mitochondria [115]. Modern endosymbiosis of photosynthetic bacteria is also known [116,117], as well as endosymbiotic capture of photosynthetic eukaryotes by other eukaryotes [118]. Endosymbiosis, consequent symbiote genome reduction [119] and prokaryote-to-eukaryote gene transfer [120,121,122] are modern as well as ancient phenomena. 

Endosymbiosis is also not confined to the eukaryotes. Modern biology shows examples of endosymbiotic or endoparasitic bacteria that live inside other bacteria [123,124], and bacteria that live inside modern mitochondria [125]. Endosymbiosis is therefore possible, even common, and is likely to have occurred many times in the history of life.

There are various mechanisms by which conversion of a symbiont or parasite to an obligate endoparasite, and thence to an organelle, can happen, for example the: (1) gradual increase in physical tightness of the association between microbes found in a consortium until the association transitions from partial surrounding of one member by the other to full engulfment; (2) the acquisition of a bacterium as food, as in a food vacuole, which then escapes to the cytoplasm of the would-be host and taking up long-term residence; or (3) by intracellular parasitism from a bacterium which transitions to a more mutually beneficial relationship [126]. Particularly, the parasitic endosymbiotic pathway seems often to be underestimated in importance, considering that even parasites themselves have endosymbionts such as *Toxoplasma gondii*, which uses a plant hormone for communication that is derived from a relict endosymbiont and which was acquired by the ingestion of a red algal cell [127].

So it seems unlikely that endosymbiosis itself was uniquely unlikely. The initial eukaryote could have rapidly dominated its ecological niche (a Pulling Up the Ladder event), or several independent ur-eukaryotic lineages could have arisen, with only the one surviving to this day. If so, a possible analogy can be drawn to the acquisition of mitochondria, which appear to be monophyletic and might be a pull-up-the-ladder event as well. However, there is a considerable biochemical diversity of anaerobic mitochondria (e.g., some use organic compounds rather than molecular oxygen as final electron acceptors under anaerobic conditions) and biochemical heterogeneity of aerobic mitochondria (some of which appear to lack a remnant genome [128,129]). Thus, the progress that mitochondria or mitochondria-like endosymbionts provide can be considered a Many Paths event, even if mitochondria themselves are not.

While endosymbiotic acquisition of mitochondria and plastids was a critical event in the origin of eukaryotes, it is not known if it was the only such critical event. Specifically, it is presently not known if the “host” cell which acquired the endosymbiont precursors of mitochondria, plastids, and potentially other organelles already had a nucleus, endoplasmic reticulum, golgi apparatus, i.e., if the cell already had some of the internal compartments that are distinctive features of the modern eukaryotic cell. 

Endosymbiosis is not the only route to developing internal compartmentalization, and such compartmentalization is not unique to eukaryotes. *Planctomycetes* are prokaryotes, but some have an intracellular membrane-bound compartment for carrying out the anammox energy-generation reaction [130]. Some archaea also have internal membrane compartments [131]. Complex internal membrane stacks are common in cyanobacteria [132]. Membrane-bound nuclear bodies are also well known in the prokaryotes (discussed in the next section). Broad statements that intracellular membranes are unique to the eukaryotes (e.g., [133]) are therefore incorrect. The intracellular membrane system of eukaryotes is integrated into a dynamic network of vesicle trafficking and control which is rare in prokaryotes (reviewed in [134]); however some of the core proteins and structural elements of a cytoskeleton are also found in prokaryotes [110,135,136,137,138,139], and the giant bacterium *Epulopiscium fishelsoni* has an internal tubule system so similar to eukaryotes that it was initially mistaken for a protozoan [140,141]. This suggests that the eukaryotic cell architecture is an elaboration of a system that was either pre-existing in the LCA or has evolved several times since in response to growth in cell size.

Whether the complex internal structure of the eukaryotic cell arose by elaboration of internal components or by endosymbiosis, both have clearly arisen many times and so are examples of Many Paths events. We contend that the uniqueness of the eukaryotic cell does not lie in its structure, but rather it lies in the eukaryotic genome, which allows complex elaboration of internal structure, which will be dealt with below.

### 5.5. Eukaryotic Gene Organization

Eukaryotes have far more elaborate gene control systems than prokaryotes, and their acquisition of this genetic complexity is seen as a key step on the way to multicellularity and complex organisms. This elaboration is intimately coupled with the eukaryotic chromosome structure, itself intimately linked to the cytoskeletal apparatus of mitosis and meiosis [133]. 

We separate the evolution of the complex genetic of eukaryotes from the transition leading to complex, multicellular organisms because: (i) eukaryotes (and hence presumably their genetic architecture) evolved before complex multicellular organisms; and (ii) the majority of classes of eukaryotes today remain single celled organisms. (We emphasise that most eukaryotes, by number and by classification, are not multicellular. They have intracellular structure that is more complex, as discussed above, but the point here is not that all eukaryotes are on the path to multicellularity, but that eukaryotes evolved features which enabled multicellularity, feature which bacteria and archaea apparently lack. One of these is a genetic architecture that allows the developmental complexity of complex, multicellular organisms).

This key innovation is discussed in detail in [32], so a brief treatment with the main points will suffice here: There are multiple types of control of gene activity in eukaryotes that overlap with each other. The different control functions have evolved many times with the same general type of genetic function often being carried out by different chemistries in different organisms. Many types of control chemistry in eukaryotes have precedent in bacteria or archaea. Thus, for example, small RNA control of RNA chemistry is found in all domains of life, and has evolved independently at least twice in eukaryotes. While the specifics of mammalian piRNA chemistry may be unique, the evolution of the regulation of gene activity through protein-mediated recognition of mRNA by small RNAs is a Many Paths process. Bains and Schulze-Makuch [32] provide an exhaustive list of other examples of all levels of nucleoprotein organization, transcription, translation, mRNA and protein breakdown, and RNA and protein chemical modification, and show that all are likely to have evolved through a Many Paths process. 

If we are to dismiss the chemical differences between bacteria and archaea on the one hand and eukaryotes on the other, how do we explain that some eukaryotes are clearly more complex than any prokaryote? Why are there not prokaryotes with as complex genomes as (say) *Caenorhabditis elegans*? We argue that there is a basal feature of eukaryotic genomes that allows this complexity, and prevents it in prokaryotes. The key feature is that the default state of eukaryotic genes packaged in chromatin is “off”. The entire nucleoprotein apparatus of eukaryotes is structured so that the DNA in it is not transcribed unless actively allowed to do so. By contrast, the nucleoprotein of bacteria and archaea is structured so that genes are readily transcribable, and control is often, perhaps commonly, through repression or absence of specific transcription factors. 

The reason that the default off state is important is as follows. Gene control systems are not computer code, no matter how the language of molecular or synthetic biology suggests otherwise. If the circuits of even a yeast’s genome were coded in conventional computer language, they would be “Spaghetti code” of the worst type, as the control levels and mechanisms interact arbitrarily with each other to produce results that are authentically chaotic. At a cell and tissue level, if the global circuitry controlling cell differentiation in mammals is analysed (rather than the role of a single effector being illuminated, and therefore claimed to be “critical”), it is found that every control system described above interacts with every other. Examples range from the mammalian development of white and brown fat [142] and neurogenesis [143,144] to yeast mating type loci control [145,146]: all of miRNA, piRNA, lncRNA, protein transcription factors, specific DNA sequence elements, and local, regional and global histone methylation and acetylation make a spaghetti code of interactions to define the biological endpoint. 

The complexity of genetic circuits is therefore not just a function of the number of coding and regulatory elements, but of the number of ways they can interact, so that the number of distinct genetic programs is a polynomial function of genetic complexity, providing that an ever-increasing number of genes and transcript types can co-exist in the nucleus. If a genetic program requires the transcription of a set of genes, then it also requires no expression of all the other genes in the genome. It was a well-known observation from the dawn of molecular genetics that most of the genome is not transcribed in most cells of a multicellular body, nor in single celled organisms most of the time. To add a new set of genes to a genome, not only must a unique control network for that gene set be created, but a way of not activating all the other genes in the genome must be implemented as well. If the default status of genes is “off”, then this second task is already achieved. If the default state of the genes is “on”, then the first task is easier, but the second requires modulation of every other gene’s control system in the organism. 

Thus we postulate that the evolution of a genome in which the default expression status was “off” was the key, and unique, transition that allowed eukaryotes to evolve the complex systems that they show today, not the evolution of any of those control systems per se. Whether the evolution of a “default off” logic was a uniquely unlikely, Random Walk event or a probable, Many Paths, event is unclear at this point [32].

### 5.6. Multicellularity

The evolution to multicellularity does not appear to be a distinct step, but is polyphyletic. Examples are colonial organisms, closely connected single-celled organisms that are interdependent such as microbialites (e.g., [147]) and facultative multicellular organisms, which deploy multicellular functions as a result of environmentally triggers [148]. There are organisms that normally spend all of their life cycle as multicellular units, but which can show no cell differentiation and can survive and reproduce as single cells. There are also organisms that spend more of their life cycle as singular cell beings, but require a brief multicellular stage at some point in their life. The question arises then what is multicellularity and where really is the critical transition?

Here we adapt the definition by Bell and Mooers [149] that multicellular organisms are “clones of cells that express different phenotypes despite having the same genotype”, but add the clarification proposed by Resendes de Sousa António and Schulze-Makuch [113] that this differentiation is cooperative and not competitive, is required for the survival of the organism, and genetically predetermined. The program for the differentiation is genetically transmitted to the next generation and the organism as a whole cannot revert back to a non-differentiated life style. This type of obligate multicellularity is only observed in eukaryotes, while facultative multicellularity and other forms of close interconnectedness is observed among species of all domains: archaea, bacteria, and eukaryotes. 

It is unclear how multicellularity evolved. Several hypotheses have been put forward, the most common being that colonial unicellular organisms evolved multicellularity when exposed to environmental stress or that it originated from a symbiotic relationship between different species of unicellular organisms (e.g., [150]). It has also been proposed that multicellularity could have arisen from accidently joined cells, particularly during the reproduction process, or by organisms, such as ciliates, that have two or more nuclei and went through a genetic split (e.g., [113]). Fairclough et al. [151] suggested that multicellularity might have evolved via post division adhesion such as displayed by choanoflaggelates, the closest unicellular ancestor to animals.

Clearly, obligate multicellularity is a conceptual shift in life strategy, because the unit of task division is within the genetic material and the organism cannot revert back to a unicellular life style. Thus, for our anthropocentric analysis the critical transition is the achievement of obligate multicellularity in which organisms cannot exchange between single-cellular and multicellular life style.

Although multicellularity and especially obligate multicellularity is a major advance toward complexity in the natural history of Earth, it seems to have evolved multiple times within the eukaryotic kingdom. Many distantly related eukaryotic branches hold multicellular life forms including Ophistokonts, Excavates, Amoebozoa, Plants, Heterokonts, Alveolates, and Discicrisstates [152]. One particularly intriguing example is the family Saccharomycetaceae, because it contains the unicellular yeast *Saccharomyces cerevisiae*, the multicellular filamentous cotton pathogen *Ashbya gossypii*, and even organisms that can switch cellularity dependent on environmental cues such as *Candida albicans*. Thus, what seems to be a critical advance toward complexity for us is easily and readily achievable in eukaryotic biology, has evolved many times independently in very different groups of organisms, many of which contain unicellular species as well as multicellular ones. Thus we conclude that the evolution of obligate multicellularity is a Many Paths process

### 5.7. The Development of Large, Complex Organisms

One of the most singular transitions of life on Earth from a human point of view has been the Cambrian Explosion. As noted above, multicellularity has arisen many times in sexually reproducing eukaryotes, but only in a few cases has this lead to large, complex organisms [153], and all of the extant lineages of such organisms appear relatively suddenly in the fossil record ~540 Ma ago. There are two broad classes of arguments concerning why complex organisms arose suddenly in the fossil record at the start of the Cambrian. The first states that geophysical conditions, and particularly the rise of atmospheric oxygen, allowed large, high-energy organisms to evolve necessitating the evolution of fossilisable hard parts. The second is that the genetics that allow complex development plans (and hence complex body plans) evolved relatively suddenly in the Vendian, allowing the Cambrian Explosion. We suggest that either mechanism is a Many Paths process. 

Most authors (e.g., [154,155]) argue that the Cambrian Explosion needed both one or more causes (pre-existing conditions) and one or more triggers to happen. We argue that both causes are Critical Path events, and so, whichever was limiting for life on Earth, the evolution of complex animals was highly likely within 1 to 2 Ga of the initial evolution of photosynthesis. We further argue that the genetic explanation is more likely than the geochemical explanation, because the Cambrian explosion was not a singular event, but a series of events over up to 500 Ma. The *Porifera* and the *Cnidaria* were well established in the Vendian, and other metazoan groups may have had primitive members there too [156] as well as the Ediacaran fauna [157]. Molecular clock data give diverse dates for the initial radiation of the Bilatera (the lineage of animals leading to all the major animal groups today), ranging from 580 Ma to 900 Ma ago for the last common ancestor of the *Cnidaria*, *Porifera* and *Bilateria* [156,158,159], all well before the Cambrian. Features once thought to have a single, Cambrian origin such as body segmentation, radial symmetry and the coelome are now understood in the light of molecular evidence to have evolved independently several times. Thus, the Vendian-Cambrian period was one of great diversification in animal body forms, but it was not a single event.

Growth and development of a complex organism requires complex coordination of gene activity in space and time. It requires that the majority of genes in a cell be turned “off”, and only the cell’s general housekeeping genes and the specific genes associated with the differentiation state of that cell be expressed. This requires a substantial expansion of the genetics involved in gene control over the genes coding for actual structural components of the cell. We argued above that this required a unique, “default off” status for the genome to allow for that complexity. Once that basic genetic logic is achieved, we argue that the development of more complex genomes, and hence more complex phenotypes, is a Many Paths process. We argue this on two grounds; that similar anatomical functions have evolved many times independently in different lineages, and that similar levels of developmental sophistication have evolved independently in different lineages. Both examples point to the independent complexification of the genome of the presumed ur-eukaryote, which suggests that such complexification was a Many Paths process.

The multiple, parallel evolution of features of anatomy or function in different metazoan lineages is well known. Examples include body segmentation [156], triploblasty [158], striated muscle [160,161], nerves [162], sexual morphology [156,163], Y chromosomes [164], imaging vision [22], and flight [165]. These point to such innovations as being the result of a Many Paths process of innovation (again, we are concerned here with the evolution of function, not necessarily with specific anatomy or biochemistry by which that function is achieved). Much of the genetics seen in all the developmental programmes that generate eyes, muscles, nerves, wings etc. uses common types of elements that have been duplicated, diverged and repurposed throughout metazoan evolution. The evolution of the many, diverse forms of the metazoa is therefore seen as an elaboration of pre-formed modules of genetic circuitry through duplication and subsequent function modification via all the diverse mechanisms summarised above. As Wilkins [156] says, “According to this point of view, the foundations of developmental evolution were laid long before there were multicellular eukaryotes. A crucial step was the evolution of molecular organizational modules involving both signal transduction and gene transcription systems. And, perhaps, with that property, the advent of multicellular complexity was a virtual inevitability”.

A deeper question therefore is whether the evolution of the modules of genetic circuitry that allow the parallel evolution of muscle, nerve, segmentation etc. is a highly improbable, Random Walk event, or whether that too could arise multiple times. We argue that it could evolve multiple times, because we have present-day evidence that it did. The underlying genetic circuitry that can (and in one lineage did) give rise to the metazoa evolved by sequential duplication and divergence of genetic controls between 700 Ma and 1000 Ma ago [166,167], building on a eukaryotic lineage going back to at least 1500 Ma ago [168]. Robust fossil evidence for multicellular animals only appears in the Vendian with the Ediacaran assemblage [157]. Metazoa probably evolved from choanoflagellate-like ancestor that contained a range of cell adhesion and signalling protein domains, which have been re-purposed through domain shuffling in metazoa [169].

Some of the underlying control systems of the metazoa are thought to have evolved at least twice. Protein tyrosine kinases for example mostly evolved independently in metazoa, choanoflagellates and ichthyosporans, although some are conserved between these eukaryotic clades [167].

Qualitatively, it is clear that complex organisms with sophisticated developmental plans have evolved several times in different lineages. Vascular plants and multicellular fungi are not usually considered comparator groups for animals, but they obviously are. Land plants have specialised metabolic, transport, sensory and surface defence tissues, multi-layered, polarized cell layers in all tissues [170], complex non-repetitive morphology [155], highly sophisticated reproductive organs and strategies, and are capable of rapid organism-wide electrochemically-based signal transduction [171,172] (although not “nerves” in a vertebrate sense [173]), rapid movement [174], directional light sensing, and metabolic thermogenesis [175]. Their development is coordinated by complex arrays of sequentially expressed genes [170]. In all regards, vascular plants are more developmentally complex than the Porifera. If the Cambrian Explosion had not had the metazoan ancestors to work with, parallel evolution in other clades such as the Plastida could have produced equivalent genetic complexity. 

We realise that this argument will not convince some. Could there exist a robust and diverse biosphere consisting of only slowly metabolizing, non-motile, but quite complex plant-like organisms? Surely we are not saying that, if animals had not been here, plants would have evolved blood, muscles and brains? However, that is exactly what the evidence suggests. If we accept that an organism with no more nerve, muscle or brain than the Ediacarans, but with the flexibility of the underlying eukaryotic genetics, could evolve into mammals, then we must accept that any equivalently complex eukaryote lineage could produce complexity equivalent to a mammal given the necessary time. One organism—such as *Kimberella quadrata* during the Precambrian on Earth [176]—will eventually develop the capability of movement and remote detection of potential prey, filling the ecological niche of “predator”. This is a pull-up the ladder event, because once that niche is filled, no phototroph as motile as *Kimerberella* can take over that niche. Given that we have multiple examples of such organisms along the path from the simplest multicellular organism to humans, then we must also accept that each of those steps is not itself extremely improbable, but that it “only” takes a long time to accumulate a suitable set of genes. It therefore follows that the path from eukaryotic genetics to complex animals is a Many Paths process, and takes in the order of a billion years to traverse. That our ancestors walked it first is an accident of history, but someone has to be first.

### 5.8. Intelligence

Intelligence can be defined as the ability to integrate experience so as to anticipate the future [177] through means other than selection (i.e., death of those that did not anticipate correctly, as operates in the immune system, for example). Operationally, it is detected as an observation of complex adaptive behaviour in natural settings, by tool use, or by the ability to learn complex non-natural behaviour (e.g., laboratory problem solving). A nervous system seems to be the prerequisite for intelligence, but once that is a given, intelligence evolved independently in several major groups of animals on Earth. Intelligence seems to go hand in hand with enlargement of the central nervous system, and in species closely related to us such as mammals exceptional enlargement of the brain, but enhanced cognitive capacity emerged from different starting points in the more intelligent species. For example, cephalopods and vertebrates show the ability to learn how to make tools (e.g., [178,179,180]), but have completely different CNS anatomy: the only commonality is that they have bigger brains than less cognitively capable sister groups, with a higher encephalization quotient. (Note, however, that the encephalization quotient is not suitable for a direct comparison between non-related types of organisms. A striking example is the octopus in which only part of its complex nervous system is localized in the brain with two-thirds of neurons found in the nerve cords of its arms.).

A number of other factors are likely to be pre-requisites for developing high intelligence (Table 2), but again these are widespread across many groups of organisms. Given that increased intelligence arose many times in vastly distantly related organisms and very different time periods, it can be concluded that the evolution of intelligence is a Many Paths process.

### 5.9. Technological Intelligence, or “Are There Visitors in the Cosmic Zoo?”

We understand technology to be something more than tool use. As mentioned above, tool use is relatively common in a wide variety of animals. However human technology differs from these examples in extent and in kind. Our technology is designed and taught (rather than instinctive), is specialized (so that some humans do nothing but make artefacts for others to use), and harnesses external power. It is also passed on by explicit teaching from generation to generation, a task that in theory does not require language but in practice uses it extensively. Only the human species *Homo sapiens sapiens* achieved significant technology capabilities on Earth. The underlying reason that no other species has developed a technology might be that technological ability requires a whole set of evolutionary advancements as summarized previously by Irwin and Schulze-Makuch [181], including a threshold of neural complexity, manual dexterity, social complexity, long life and post-reproductive survival. There were certainly other human species that were able to manufacture tools and used them in broad contexts such as *Homo ergaster*, *Homo habilis*, and *Homo erectus*, with some members of the *Homo* species likely being able to control fire (e.g., *Homo heidelbergensis* and *Homo neanderthalensis*). Are the achievements of the other human species still on the level of tool manufacture and use as observed in other animal species, or did some of them already reach the threshold of what we would call technological intelligence? Would the control of fire be that threshold or perhaps only that ability in conjunction with manual dexterity, complexification of social interactions and particularly the development and dependence of a language as seen in *Homo sapiens sapiens*? If so, the evolution of these capabilities in the genus Homo would represent a Critical Path process, a highly likely development from in initiating event which separates *Homo* from *Pan* and *Gorilla*. In this case we cannot consider other *Homo* species as being independent examples of technological intelligence. 

Thus we cannot say with certainty whether the evolution of technological intelligence is a Critical Path process, a Many Paths process or even a Random walk event (which we consider much less likely but cannot exclude at this time), and we cannot say whether a fraction of 2nd Earths elsewhere would eventually develop technological intelligence. The likelihood of visitors, the evolution of technological intelligence, cannot be evaluated at this time. Based on Earth’s natural history and the evolutionary advancements needed for technological intelligence, particularly the requirement of stable time to evolve a complex social structure, it might indeed be rare. 

## 6. Discussion

It is beyond the scope of this paper to address all of the substantial literature on the nature of the selection that leads to the various key innovations, although we note that very general considerations suggest that organisms that can build phenotypes that can adapt to a wide range of environments, in space or time, are likely to be at an advantage over organisms with simpler phenotypes than can fit only a few environments [182]. Our approach explicitly does not try to unpick either the specifics of the selective forces or the detailed evolutionary mechanisms by which features evolved. We wish to understand not how complex life arises, but the chance that it arises. Instead of focusing on evolutionary mechanisms and selection [183], we use models based on how often events happen (which is at least observable), specifically the Critical Path, the Random Walk, the Many Paths, and The Pulling Up the Ladder model, which assume that such selection exists. We are empirically seeking to understand the number and statistical nature of the preconditions for the key innovations. To take a specific and very extensively explored example, there is a wide range of theories on why and how multicellularity evolves (e.g., [113,148,149,152]). Regardless of path and for whatever reason, multicellularity clearly does evolve, and has done so many times relatively fast given similar environmental conditions, habitat heterogeneity, and sufficient biodiversity and biomass, as discussed above. Thus, our conclusion is that a Random Walk model does not apply to the evolution of multicellularity, but the Many Paths model does apply. A similar approach is applied to the other key innovations.

We did not deal with all innovations, which other authors considered important. One of these examples may be the invention of sex. Clearly sex is essential for most eukaryotes, and is related to the eukaryotic replication mechanism. Historically it has been argued that sex is central to the evolution of complex forms, because it allows for rapid genome evolution and the accumulation of divergent genes through diploidy, as the basis for gene duplication and divergence of function. More recently this view has been challenged, and sex is seen as a restraint on over-zealous purifying selection acting on different stages of complex life cycles [184] or a process that provides the genetic variation necessary for compensatory co-adaptation of the cellular nucleus to keep pace with the accumulation of mitochondrial mutations [185]. Either way, sex is not the only mechanism for genome renewal and new gene acquisition in animals. A range of animals have been shown to be completely asexual today, and probably asexual for many millions of years [186]. Bdellid rotifers have been asexual for ~60 Ma [187], and have a genome that is definitely incompatible with meiosis. Instead, they seem to acquire new genes by non-meiotic horizontal gene transfer from many organisms, including prokaryotes [188]. Oribatid mites probably originated in the Silurian and have radiated to more than 10,000 species, about 10% of which reproduce parthenogenetically [189]. Thus the evolution of complex animals and sex are not obligatorily linked. 

The evolution of sex is also distinct from the evolution of soma/germ-line separation. Plant evolution shows that soma/germ-line separation is not essential for the development of complex multicellular organisms [190]. Many flowering, sexually reproducing plants can regenerate whole plants (including germ-line sexual cells) from tiny fragments of somatic tissue. Land vertebrates have lost this capability, but it is widespread among ancestral invertebrates such as the *Cnidaria* and *Porifera*. It is important from our point of view that protein components of the meiotic apparatus also show great diversity across these species. Centromeric sequence and proteins evolve rapidly [191], including centromeric variants of otherwise highly conserved chromatin proteins such as histone H3 [192]. Thus, we interpret sex not as a key innovation, but a phenomenon “that just happened” among other possibilities to achieve the same outcome. 

Certainly, it is somewhat arbitrary and in the view of the beholder (as human species) what we consider a key innovation or transition of life. A comparison of these with the ones identified by our peers is provided in Table 1. 

Based on our analysis, all key innovations identified by us can be attributed to a Critical Path or a Many Paths process with the exception of the origin of life and possibly the rise of oxygenesis and of technological intelligence. This means that if environmental conditions are conducive on a planetary body for long enough time, biological complexity is very likely to arise once the origin of life has occurred (e.g., [41]). In regard to the origin of life data are lacking to attribute it to a specific process. The same is the case for the rise of technological intelligence, because it occurred only once on our planet during its natural history of 4.6 Ga. 

In essence, the result would be that we either have many planetary bodies in the galaxy and the universe as a whole, where biological complexity and at least animal-type life is abundant or—if the origin of life is extremely rare—we live in a (biologically) rather empty universe. 

## 7. Conclusions

After establishing a set of preconditions needed for life, which are similar to the criteria for habitability that have been discussed extensively in the astrobiology literature, we examined the key innovations of life on Earth, and tested them for multiple occurrences. Using a consistent approach and a model toolset we find that, with the exception of the origin of life and the origin of technological intelligence, we can favour the Critical Path model or the Many Paths model in most cases. The origin of oxygenesis, may be a Many Paths process, and we favour that interpretation, but may also be Random Walk events. This implies that in any world where life has arisen and sufficient energy flux exists, we are confident that we will find complex, animal-like life. The example of Earth suggests that such complex life could evolve in a 1–10 Ga timescale. This does not mean that, as Conway-Morris argues [21], life has to end with humans or animals we are familiar with, but suggests that if we rewind the tape, the result will be same function, but different anatomy and very possibly also different chemistry.

Our conclusion has implications for the search for life on other worlds. Not only should we expect microbial biosignatures, but also the detection of signatures that depend on large, complex, multicellular organisms such as the vegetation Red Edge (if it is relevant to the physics and chemistry of the world—see [39]). In particular, this is relevant to the selection of tools we use in searching for life on planets in other solar systems. We hope that our analysis will give support to the search for such life. 

## Figures and Tables

**Table 1 life-06-00025-t001:** Key innovations towards humanity.

Key Innovation	Sub-Category of Innovation	References					
		A	B	C	D	E	F
The origin of life		•					•
Photosynthesis		•					
Oxygenesis		•			•		•
Extremophily					•		
Eukaryotic cell organization		•	•	•	•	•	•
	Gene organization	•					
	Endosymbiont acquisition	•			•		
			•	•		•	•
Multicellularity		•			•		
	Cell/organism differentiation	•	•			•	
Plant colonization of land				•			
Animals evolution		•		•			
	Movement						•
	Sight						•
	Homeothermy			•			•
	Nervous systems			•		•	
Intelligence		•					
Consciousness *							•
Human society/language/technology		•	•	•	•	•	

Summary of commonly discussed key innovations on the path to complex organisms. References: A: this study; B: [1]; C: [23]; D: [24]. E: [25]; and F: [26]; Some of these authors subdivide the innovations listed here; for consistency, we do not list these subdivisions except for the animals, for historical context. * We note that some authors list the acquisition of consciousness as a key innovation. We do not include consciousness as such as a key innovation in our study, in part because it is intimately tied up with intelligence but mainly because neither scientists nor philosophers can decide how to determine whether an organism has consciousness or not.

**Table 2 life-06-00025-t002:** Factors promoting the evolution of intelligence.

Promoter	Reasoning	Example
Body Size	Animals with individual intelligence tend to have large body sizes compared to the average of their taxonomic group	Cephalopods, elephants
Activity Level	Active organism that move through changing environments are required to analyse features that—sedentary organisms do not have to deal with, like acceleration and balance, depth perception, feature extraction, distinguishing foreground from back-ground, etc.	Dolphins, humans
High Sensory Resolution	High sensory processing, such as visual and tactile, requires high intelligence, for example in complex arboreal environments	Primates, parrots,
Fine Motor Control	Fine motor control is needed for a combination of delicate and complex movements including the coordination of multiple appendices and subtle muscle movement to control vocalization	Octopus, parrots, human
Social Behaviour	Intelligence is required for sophisticated communication either by behaviour, vocalization, or facial expression. It often involves hierarchical and territorial awareness, and accurate social memory	Primates, cetaceans, social insects *

Modified from [177]. * Meta-intelligence, not individual intelligence.

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
