# Peer review of "The Cosmic Zoo: The (Near) Inevitability of the Evolution of Complex, Macroscopic Life"

_life, 2016, doi:10.3390/life6030025_

Round 1

Reviewer 1 Report

This article is of high interest for a broad readership, thus not for scientists especially, since the question of existence of ETs has got high actuality given the present reports of exoplanets. The authors identify major steps or innovations along the course of the evolution of life. It seems that the authors take it for granted that this course is characterized as a steady increase of complexity.  However, such a view is at odds with the common scientific paradigm, inasmuch as increasing complexity is not clearly upheld by most evolutionary biologists – a circumstance that I think the authors should mention and discuss. Such discussions of increasing complexity are to be found in

Ekstig, B. 2016 B. Mechanisms of Progress in Organic and Cultural Evolution. International Journal of Social Science Studies Vol. 4, No. 2. 

Lineweaver, C. H., Davies, P., and Ruse, M. 2013. (Eds.) Complexity and the Arrow of Time. Cambridge: Cambridge Univ. Press.

Author Response

Reviewer 1.

This article is of high interest for a broad readership, thus not for scientists especially, since the question of existence of ETs has got high actuality given the present reports of exoplanets. The authors identify major steps or innovations along the course of the evolution of life. It seems that the authors take it for granted that this course is characterized as a steady increase of complexity.  However, such a view is at odds with the common scientific paradigm, inasmuch as increasing complexity is not clearly upheld by most evolutionary biologists – a circumstance that I think the authors should mention and discuss. Such discussions of increasing complexity are to be found in

Ekstig, B. 2016 B. Mechanisms of Progress in Organic and Cultural Evolution. International Journal of Social Science Studies Vol. 4, No. 2.  

Lineweaver, C. H., Davies, P., and Ruse, M. 2013. (Eds.) Complexity and the Arrow of Time. Cambridge: Cambridge Univ. Press.

  We are grateful for this encouraging comments of this reviewer. We understand that the paper as originally written gave the impression that life always increases in complexity. We mentioned the ‘diffusion from the wall’ explanation for why some life increases in complexity with time; in the revised paper we have specifically expanded this discussion to state explicitly that we are concerned with the development of more complex forms, and so focus on that, however the majority of the biosphere remains ‘simple’, and many examples of decreasing complexity are also found. Specifically, we are not trying to defend the ‘evolution as progress’ argument. We are grateful for bringing Lineweaver et al to our attention: Ekstig addresses a tangential point in our view, and we have not cited this paper. 

Reviewer 2 Report

GENERAL ASSESSMENT

This manuscript argues in favor of the hypothesis that evolution of complex life is highly likely in any stable, sufficiently large environment where there is life already. The authors review some facts and speculations about a set of fundamental evolutionary transitions that living systems on Earth have gone through and, extrapolating from that reconstructed and partial history of events, defend their hypothesis.

The contents of this work, if circumscribed to their review part, could have relative interest. However, the main thesis proposed in it is excessively ambitious and, more importantly, the method that the authors apply to support it is not tenable (see my critical remarks below). Therefore, I recommend rejecting the paper.           

MAIN CRITICAL REMARKS
All biological phenomena are highly intricate, difficult to fully account for, because they rely on systems of enormous diversity, both in terms of molecular/cellular composition and in terms of dynamic interactions among those components. In addition to that (and partly also because of that), it is extremely difficult to know how biological processes depend on their history -- e.g., whether future dynamic behaviors/phenotypes are somehow restricted or, rather, enabled by previous states and conditions. The complex (non linear and multiple) intertwining of historical and organizational constraints is precisely one of the features that make living systems, especially when studied as a whole (i.e., as complete cells or multicellular entities) so tricky and unpredictable. And that is why biology has remained essentially a descriptive activity over the years -- in contrast with other natural sciences, like physics or chemistry.

Given this deficit in understanding and predictive power, not only apparent in the context of life’s origins (as the authors seem to be happy acknowledge) but also thereafter (and more acutely in relation to events like major evolutionary transitions), one might be tempted to take a retrospective approach and try to extract some conclusions from historical data.  This is the general strategy followed in this paper. Nevertheless, the authors of the manuscript do not seem to be aware that the claims that one can make from the analysis of historical data are also strongly limited. Testing a hypothesis like the one they propose (the near-inevitability of complex life) through such a retrospective approach is doomed to failure.

More precisely, the way of arguing in the paper is unsound (if not totally flawed), because of the following points:

1) Historical data have an inherent contingent nature. Otherwise they are not considered historical.  And what can be extracted from contingent events, in relation to the probability of a later --or a parallel, independent-- occurrence of a similar event? Nothing. Simply nothing.   

2) If one knows the probabilities of certain events and the final outcome of a process, even if multi-causal steps are considered, one could in principle determine retrospectively what was the path followed (e.g., by Bayesian inference rules), but in order to do so quite precise knowledge about the system (e.g., the relevant parts and interactions involved at each step, the corresponding conditional probabilities, etc.) is required. Nothing on those lines is carried out in the paper. No effort of disentangling the historical (contingent) from the necessary, organizational or systemic constraints is made; and without that delimitation work, without that preliminary sorting analysis, all subsequent reasoning goes to waste.

3)  Instead, the authors present a set of 3-4 transition models, of their own making, through which they claim (without much discussion or proof) that all types of possible transitions are covered: the 'critical path model', the 'random walk model', the 'many paths model', plus the 'pulling-up the ladder' one. Then, through a hardly falsifiable analysis, they review different transitions that took place in the history of life on Earth, and classify those transitions according to the models proposed. "Of course", the desired conclusions follow.

4) The revision of the 'state of the art' regarding major transitions in evolution is not deep, nor complete enough. The authors are working with partial historical reconstructions, what makes their goal even less accessible. This becomes particularly apparent for the case of the emergence of eukaryotes. The fact that endosymbiosis (stated in such general terms -- without going into the details of each case) is not confined to the eukaryotic transition says absolutely nothing of relevance to establish whether that transition was probable or not in the first place, (or whether that would be more or less probable if another biosphere was generated on an alternative planet). Simply, we haven't got a clue. Similarly, reducing the complexity of eukaryotic genome organization to "the fact" that the default state of the corresponding genes is 'off' is an oversimplification with no proper scientific grounding. Methylation/demethylation patterns are much more complicated, with huge differences between plants, fungi and animals, even between vertebrates and invertebrates. As an example (or counter example), DNA methylation is not present in the testes of all mammals, so all their genes would be ‘on’ by default.

5) Tinkering (a concept that, surprisingly, is not mentioned in the paper, not even once) is pervasive in the history of life on Earth, as Jacob highlighted long ago. Living beings construct from what has been already constructed, use similar strategies and tricks, if they happen to be available. They do not design everything 'de novo' -- quite the opposite. Therefore, many correlations like the ones drawn in this manuscript could be made. But this does not mean that major evolutionary transitions are more or less probable/inevitable, more or less difficult to occur. One has to go step by step and study in depth all the relevant aspects involved in each transition. Then we might be able to generate more reasonable estimates or extrapolations. However, the knowledge that we need to acquire in order to carry out the proper calculations required to support the claim of the authors of this paper is simply not available at present… and who knows whether it will ever be. Let us be hopeful, anyway -- but in the meantime, more cautious. 

Author Response

Reviewer 2.

This manuscript argues in favor of the hypothesis that evolution of complex life is highly likely in any stable, sufficiently large environment where there is life already. The authors review some facts and speculations about a set of fundamental evolutionary transitions that living systems on Earth have gone through and, extrapolating from that reconstructed and partial history of events, defend their hypothesis.

The contents of this work, if circumscribed to their review part, could have relative interest. However, the main thesis proposed in it is excessively ambitious and, more importantly, the method that the authors apply to support it is not tenable (see my critical remarks below). Therefore, I recommend rejecting the paper.

We understand the referee’s comments. Our intention is to provide a review in a framework, rather than just a list of facts. This was meant to be ambitious – we regret that this reviewer thought we have over-reached ourselves. We hope the revisions make the paper more acceptable.

MAIN CRITICAL REMARKS
All biological phenomena are highly intricate, difficult to fully account for, because they rely on systems of enormous diversity, both in terms of molecular/cellular composition and in terms of dynamic interactions among those components. In addition to that (and partly also
 because of that), it is extremely difficult to know how biological processes depend on their history -- e.g., whether future dynamic behaviors/phenotypes are somehow restricted or, rather, enabled by previous states and conditions. The complex (non linear and multiple) intertwining of historical and organizational constraints is precisely one of the features that make living systems, especially when studied as a whole (i.e., as complete cells or multicellular entities) so tricky and unpredictable. And that is why biology has remained essentially a descriptive activity over the years -- in contrast with other natural sciences, like physics or chemistry.

Exactly so, and we are delighted with this comment. This is why we do not try to detail every hypothesised evolutionary path for every feature of every complex organism on Earth. Rather, we have sought to cut the Gordian knot and ask how often the result of that path is seen. If rarely, then it is rare. If often, then it is common. Only on a couple of occasions (notably in the discussion of the evolution of oxygenesis) have we opened up the can of worms that is hypothesised evolutionary mechanism, for reasons we have explained in that section. We have tried to be much more explicit about this approach in the paper.

Given this deficit in understanding and predictive power, not only apparent in the context of life’s origins (as the authors seem to be happy acknowledge) but also thereafter (and more acutely in relation to events like major evolutionary transitions), one might be tempted to take a retrospective approach and try to extract some conclusions from historical data.  This is the general strategy followed in this paper. Nevertheless, the authors of the manuscript do not seem to be aware that the claims that one can make from the analysis of historical data are also strongly limited. Testing a hypothesis like the one they propose (the near-inevitability of complex life) through such a retrospective approach is doomed to failure.

The reviewer makes two points here. Firstly, that the claims that we can make from historical analysis of this sort are limited. We agree, of course. This is not a deterministic analysis of a physical system. As to testing the hypothesis, we agree that this is hard. Finding complex life elsewhere is going to be astonishingly hard. Perhaps our conclusion is that it is worthwhile trying. Again, we have tried to expand this point to be more explicit

More precisely, the way of arguing in the paper is unsound (if not totally flawed), because of the following points:

1) Historical data have an inherent contingent nature. Otherwise they are not considered historical.  And what can be extracted from contingent events, in relation to the probability of a later --or a parallel, independent-- occurrence of a similar event? Nothing. Simply nothing.   

We regret that we did not make our point clear here. Of course the nature of a specific historical path is contingent. We argue that if several different paths result in the same result, then that result is more likely to be the result of a different path than if we only see the result once. A thrown dice never follows the same path through space, but 1/6 times will show a ‘6’ – we can estimate the probability of a ‘6’ simply by observing throws of the dice, we do not have to simulate its dynamics. We are not concerned here with the path taken but with the result, from which, with respect, we can say something. We have tried to explain this by expanding this part of the paper significantly.

2) If one knows the probabilities of certain events and the final outcome of a process, even if multi-causal steps are considered, one could in principle determine retrospectively what was the path followed (e.g., by Bayesian inference rules), but in order to do so quite precise knowledge about the system (e.g., the relevant parts and interactions involved at each step, the corresponding conditional probabilities, etc.) is required. Nothing on those lines is carried out in the paper. No effort of disentangling the historical (contingent) from the necessary, organizational or systemic constraints is made; and without that delimitation work, without that preliminary sorting analysis, all subsequent reasoning goes to waste.

 Again, we are concerned with output, not process. We have tried to make this clear.

3)  Instead, the authors present a set of 3-4 transition models, of their own making, through which they claim (without much discussion or proof) that all types of possible transitions are covered: the 'critical path model', the 'random walk model', the 'many paths model', plus the 'pulling-up the ladder' one.

We are not quite sure why the reviewer considers our choice of models so arbitrary. It seems a logical necessity that an event with can either have no random component (Critical path), one random component (or equivalently in terms of statistics, a series of random components that have to happen in a defined order) (Random Walk), or more than one random component that can occur in an arbitrary order. Either the transition does not affect the preconditions for that transition, or it does affect them (‘Pulling up the ladder’). We did not elaborate extensively on this as we have done so before in another paper – if the editor believes that we should add further elaboration to an already long paper, we can of course do so. 

Then, through a hardly falsifiable analysis, they review different transitions that took place in the history of life on Earth, and classify those transitions according to the models proposed. "Of course", the desired conclusions follow.

The conclusions do not follow as tautologically as the reviewer suggests, and we have tried to make this clear. Four transitions cannot be well classified – the origin of life, the evolution of technological intelligence, and to a smaller degree the evolution of oxygenesis and the evolution of the eukaryotic genetic architecture (of which more below). We have tried to make it clearer where the weaknesses in our argument lie in these topics.

4) The revision of the 'state of the art' regarding major transitions in evolution is not deep, nor complete enough. The authors are working with partial historical reconstructions, what makes their goal even less accessible. This becomes particularly apparent for the case of the emergence of eukaryotes. The fact that endosymbiosis (stated in such general terms -- without going into the details of each case) is not confined to the eukaryotic transition says absolutely nothing of relevance to establish whether that transition was probable or not in the first place, (or whether that would be more or less probable if another biosphere was generated on an alternative planet). Simply, we haven't got a clue.

We disagree with this point, for reasons we had hoped that the paper made clear. If endosymbiosis is common, then it is probable. That is almost a tautology. If a key event in eukaryogenesis is endosymbiosis, then that key event is probable. Internal membrane systems are common across many diverse clades. If they are common, they are probable. Thus whether the internal structure of the eukaryotic cell derives from elaboration of internal membrane systems, endosymbiosis, or a combination fo the two, it is likely not to be a uniquely rare event.  We could elaborate each of the examples extensively, but that is to fall into the ‘Just So’ trap of trying to explain a historical event in terms of the specifics of that event. We have tried to make our reasoning clearer here.

Similarly, reducing the complexity of eukaryotic genome organization to "the fact" that the default state of the corresponding genes is 'off' is an oversimplification with no proper scientific grounding. Methylation/demethylation patterns are much more complicated, with huge differences between plants, fungi and animals, even between vertebrates and invertebrates. As an example (or counter example), DNA methylation is not present in the testes of all mammals, so all their genes would be ‘on’ by default.

We are not sure why the referee focuses solely on methylation in this comment. We have gone into the complexities of eukaryotic genetics in a separate paper. We did not wish to implant that 10,000 words of analysis here as it is already available in the open access literature, as cited in the paper. We did not wish to imply that methylation was the ‘default off’ mechanism. Our hypothesis is that nucleoprotein structure as a whole is ‘default off’ in eukaryotes – DNA methylation is one aspect of this, but RNA-mediated structure, histone methylation and acetylation, and a while range of protein factors are others. We have expanded this point slightly in the revised paper, but do not want to just repeat what is available to read (and, of course, to disagree with) elsewhere.

5) Tinkering (a concept that, surprisingly, is not mentioned in the paper, not even once) is pervasive in the history of life on Earth, as Jacob highlighted long ago. Living beings construct from what has been already constructed, use similar strategies and tricks, if they happen to be available. They do not design everything 'de novo' -- quite the opposite. Therefore, many correlations like the ones drawn in this manuscript could be made. But this does not mean that major evolutionary transitions are more or less probable/inevitable, more or less difficult to occur. One has to go step by step and study in depth all the relevant aspects involved in each transition. Then we might be able to generate more reasonable estimates or extrapolations. However, the knowledge that we need to acquire in order to carry out the proper calculations required to support the claim of the authors of this paper is simply not available at present… and who knows whether it will ever be. Let us be hopeful, anyway -- but in the meantime, more cautious. 

Again, the mechanism of achieving a function is less important than the function, and so the ‘Evolution as Tinkerer’ concept is not central to our argument, but is it is an important reminder of the contingency of evolution. We welcome the reviewer reminding us to the ‘evolution as a tinkerer’ concept, which we had omitted and have now included.

Reviewer 3 Report

I have reviewed the Bain & Schulze-Makuch paper and found it to be well written and cited a wide range of literature. The authors took on a large challenge, to characterize the models for emergence of key innovations on the way to multicelluar life with intelligence/technology. The models applied were critical path, many paths, random walk & pulling up the ladder. The authors concluded that most of the key innovations arose through critical path or many paths and therefore that given sufficient time, energy sources and stability that any planet where life arose it would inevitably result in complex life.

As to the suitability of the article for Life, I have a couple of questions.  Does Life deal with SETI, complex, intelligent life, or is the focus more on the emergence of life in different environments. How "far up the ladder" does Life deal in the evolution of life in the universe? If Life is willing to entertain speculation around the arising of complex, possibly intelligent, technologically endowed life then the article is a very good review of the issues and an informed speculation. If Life is focused on "lower rungs" about the conditions for and mechanisms operating in the emergence and early evolution of life in the universe then the article is outside our scope.

I certainly learned a lot from the article and this informed my own wider considerations of the likelihood of complex life. If I did a review I would challenge some of their statements on the order of appearance of function in their brief section on the origin of life. Also I don't think that the authors considered the powerful effects of geological/climatological insults. Planetary "snowball" events or large impacts are just two such insults that have the potential to reset the progress of biological evolution, perhaps rendering no pathway forward. In the case of Mars, the loss of atmosphere and hydrosphere would have rendered all life into microbial form deep in the crust, if it is there at all. On the other hand, such insults may have been required to allow the next phase of innovations, for a new population (i.e.: mammals) to take over from a dominant form. So intelligence, and even perhaps multicellularity may be dependent upon such rare/chance events happening at the right moments. A lot of this is covered by Bob Hazen in his popular writings.

One minor negative is the somewhat casual use of language toward the beginning, making it sound somewhat folksy. Example: "we are unashamedly interested in what leads to us, not to E.coli."

Life EISSN 2075-1729 Published by MDPI AG, Basel, Switzerland RSS E-Mail Table of Contents Alert
Back to Top